# Inverted Red Quantum Dot Light-Emitting Diodes with ZnO Nanoparticles Synthesized Using Zinc Acetate Dihydrate and Potassium Hydroxide in Open and Closed Systems

**DOI:** 10.3390/mi15111297

**Published:** 2024-10-25

**Authors:** Se-Hoon Jang, Go-Eun Kim, Sang-Uk Byun, Kyoung-Ho Lee, Dae-Gyu Moon

**Affiliations:** Department of Electronic Materials, Device, and Equipment Engineering, Soonchunhyang University, Asan-si 31538, Republic of Korea; jangk96@sch.ac.kr (S.-H.J.); tkfkdrhdms153@sch.ac.kr (G.-E.K.); 20227083@sch.ac.kr (S.-U.B.); khlee@sch.ac.kr (K.-H.L.)

**Keywords:** QD, QLED, ZnO nanoparticles, inverted structure, electron transport layer

## Abstract

We developed inverted red quantum dot light-emitting diodes (QLEDs) with ZnO nanoparticles synthesized in open and closed systems. Wurtzite-structured ZnO nanoparticles were synthesized using potassium hydroxide and zinc acetate dihydrate at various temperatures in the open and closed systems. The particle size increases with increasing synthesis temperature. The ZnO nanoparticles synthesized at 50, 60, and 70 °C in the closed system have an average particle size of 3.2, 4.0, and 5.4 nm, respectively. The particle size is larger in the open system compared to the closed system as the methanol solvent evaporates during the synthesis process. The surface defect-induced emission in ZnO nanoparticles shifts to a longer wavelength and the emission intensity decreases as the synthesis temperature increases. The inverted red QLEDs were fabricated with a synthesized ZnO nanoparticle electron transport layer. The driving voltage of the inverted QLEDs decreases as the synthesis temperature increases. The current efficiency is higher in the inverted red QLEDs with the ZnO nanoparticles synthesized in the closed system compared to the devices with the nanoparticles synthesized in the open system. The device with the ZnO nanoparticles synthesized at 60 °C in the closed system exhibits the maximum current efficiency of 5.8 cd/A.

## 1. Introduction

Quantum dot light-emitting devices (QLEDs) have attracted much attention during the past few decades because they can provide many advantages, such as high color purity, widely tunable emission, a wide viewing angle, and a fast response in displays and lighting applications [1,2,3,4]. The performance of QLEDs has been significantly enhanced by using an oxide nanoparticle electron transport layer [5,6,7,8,9,10]. In particular, a ZnO nanoparticle electron transport layer provides superior performance compared to other inorganic nanoparticles such as TiO_2_, SnO_2_, and ZrO_2_ [11,12,13,14]. Several authors have demonstrated highly efficient QLEDs exhibiting an external quantum efficiency of over 20% using ZnO nanoparticles [15,16,17]. The high efficiency of these QLEDs is due to the efficient injection of electrons into the QDs and the high electron mobility of ZnO nanoparticles. The conduction band minimum energy level of ZnO nanoparticles is about 3.8–4.0 eV [12,15]. This deep conduction band minimum energy level provides a low energy barrier for electron injection from the cathode to the ZnO nanoparticles. In addition, ZnO nanoparticles result in a low electron injection barrier at the interfaces between the QDs and ZnO nanoparticle layer so that efficient electron injection into the QD layer can occur in the QLEDs [4]. The electron mobility of ZnO nanoparticles is 10^−4^–10^−3^ cm^2^/Vs [12,18]. Thus, the injected electrons from the cathode can easily transport to the QD layer through the ZnO nanoparticle electron transport layer.

QLEDs typically consist of organic hole transport layers, a QD emission layer, and a ZnO nanoparticle electron transport layer. Since the QDs are dispersed in organic solvents such as octane, the underlying organic hole transport layer may be damaged by the organic solvents during the solution processing of QDs in normal-structure QLEDs [19,20]. Hence, solution-processable polymer materials such as poly(9-vinylcarbazole) have been typically used as a hole transport layer in normal QLEDs. The highest occupied molecular orbital (HOMO) energy levels of these polymers are typically shallow, whereas the valence band energy levels of QDs are relatively deep [2,20]. Therefore, the holes are accumulated in the hole transport layer adjacent to the interface between the hole transport layer and QD layer due to the inefficient injection of holes from the hole transport layer into the QD layer [21]. In addition, the hole mobility of the organic hole transporting materials is much smaller than the electron mobility of ZnO nanoparticles [14]. These energy level and mobility mismatches between the ZnO nanoparticle electron transport layer and the organic hole transport layer result in excess electrons in the QD layer, degrading device performance due to unwanted and non-radiative recombination processes such as electron leakage into the organic hole transport layer, nonradiative Auger recombination, and exciton quenching [22]. Therefore, inverted-structure QLEDs have been proposed to avoid solvent-induced damage to the organic layer and to enhance the electron–hole balance [23,24,25,26,27]. Small-molecule hole transporting materials with deep HOMO energy levels can be widely used in inverted QLEDs because the hole transport layer can be formed by vacuum evaporation on the solution-processed QD layer. Various approaches, such as a thin insulating layer and Mg-doped ZnO nanoparticles, have been suggested to enhance the electron–hole balance in inverted QLEDs [28,29,30,31,32]. However, device fabrication processes can be more complicated with these methods.

In this paper, we synthesized ZnO nanoparticles using potassium hydroxide and zinc acetate dihydrate in open and closed systems. The morphological, electrical, and optical properties of ZnO nanoparticles are strongly dependent on the synthesis conditions, which eventually affect the charge balance and resulting device performance. Hence, we investigated the synthesis conditions of ZnO nanoparticles required for improving the charge balance of inverted QLEDs without modification of the device structure and fabrication process.

## 2. Materials and Methods

ZnO nanoparticles were synthesized using potassium hydroxide and zinc acetate dihydrate. 0.03 mol of potassium hydroxide and 0.01 mol of zinc acetate dihydrate were dissolved in 65 mL and 125 mL methanol, respectively, under stirring. Potassium hydroxide solution was inserted slowly into the zinc acetate dihydrate solution on a heating plate. The mixed solution was stirred for 5–48 h at 50–70 °C in open and closed systems to generate ZnO nanocrystals. For the open system, the reaction flask remained open to evaporate the methanol solvent during the synthesis process. On the other hand, the reaction flask was closed with a stopper to suppress solvent evaporation in the closed system. After finishing the reaction, the precipitates were collected by using a centrifuge. After this, chloroform and a 2-ethoyethanol solution were used to redisperse the collected ZnO nanoparticles. The crystal structure of ZnO nanoparticles was investigated by the X-Ray diffraction method (Miniflex 600, Rigaku, Tokyo, Japan). The particle size and shape of the ZnO nanoparticles were investigated by transmission electron microscopy (TEM, JEM-ARM200F, JEOL, Tokyo, Japan).

Indium tin oxide (ITO)-coated glass substrates with a sheet resistance of about 10 Ω/square were used to fabricate the inverted QLEDs. The standard photolithography process was used to prepare the ITO cathode electrode patterns. Acetone, methanol, isopropyl alcohol, and deionized water were used to clean the patterned ITO substrates followed by exposing them to oxygen plasma to remove organic residue on the ITO cathode. And then, a 30 nm thick ZnO nanoparticle electron transport layer was spin-coated. After coating of the ZnO nanoparticle layer, a 15 nm thick CdSe/ZnS red QD emission layer was spin-coated. The CdSe/ZnS QDs dispersed in octane solvent were purchased from Global ZEUS (ZEUS Co., Ltd., Hwaseong-si, Republic of Korea). 4,4′-bis(9-carbazolyl) biphenyl (CBP) and MoO_3_ were used to prepare a hole transport layer and a hole injection layer, respectively. Next, 30 nm thick CBP and 10 nm thick MoO_3_ layers were vacuum-evaporated at a base pressure of 1 × 10^−6^ Torr. Thereafter, a 100 nm thick Al anode layer was deposited by a vacuum evaporation method. A shadow mask was used to define Al anode patterns during the evaporation process. The fabricated ITO/ZnO nanoparticle (30 nm)/QDs (15 nm)/CBP (30 nm)/MoO_3_ (10 nm)/Al device was encapsulated in a N_2_ globe box. The emitting area of the inverted QLEDs was 2 × 2 mm^2^. The current density and luminance of the inverted QLEDs were measured by using source-meters (Keithley 2400, Solon, OH, USA) and a calibrated Si photodiode. A spectrophotometer (Minolta CS1000, Tokyo, Japan) was used to measure the photoluminescence (PL) and electroluminescence (EL) characteristics of the ZnO nanoparticles, red QDs, and inverted QLEDs.

## 3. Results and Discussion

ZnO nanoparticles were synthesized using potassium hydroxide and zinc acetate dihydrate at 50, 60, and 70 °C for 24 h in open and closed systems. A total of 0.03 mol potassium hydroxide dissolved in methanol solvent and 0.01 mol zinc acetate dihydrate in methanol solvent were used for synthesizing the ZnO nanoparticles. For the open system, the stopper of the reaction flask was opened for evaporating the methanol solvent during the reaction time. On the other hand, the stopper was closed to minimize the evaporation of the solvent in the closed system. The X-Ray diffraction patterns of the ZnO nanoparticles synthesized at 50, 60, and 70 °C in the open and closed systems are shown in Figure 1. The X-Ray diffraction patterns exhibit the characteristic peaks of the ZnO hexagonal wurtzite structure in the (100), (002), (101), (102), (110), (103), (112) planes. The X-Ray diffraction pattern of the ZnO nanoparticles synthesized at 50 °C in the open system exhibits relatively broad peaks compared with the ZnO nanoparticles synthesized at a higher temperature, indicating that the ZnO nanoparticles synthesized at 50 °C have a smaller particle size than the nanoparticles synthesized at a higher temperature. The ZnO nanoparticles synthesized in the closed system also exhibit similar X-Ray diffraction patterns, where the diffraction peaks become slightly broader as the synthesis temperature decreases.

TEM images of the ZnO nanoparticles synthesized at 50, 60, and 70 °C in the open and closed systems are shown in Figure 2. From the images, the synthesized ZnO nanoparticles are approximately spherical in shape regardless of the synthesis temperature and the open and closed systems. Comparing Figure 2a,c, the particle size of the ZnO nanoparticles synthesized at 50 °C is smaller than that of the nanoparticles synthesized at 70 °C in the open system. The ZnO nanoparticles synthesized in the closed system also show that a higher synthesis temperature results in a larger particle size. In addition, the ZnO nanoparticles synthesized at 50, 60, and 70 °C in the open system are larger than the nanoparticles synthesized at the same temperatures in the closed system. Since the solvent evaporates in the open system during the synthesis process, this result means that the concentration of the solute affects the particle size.

Figure 3 shows the particle size distributions of the ZnO NPs synthesized at 50, 60, and 70 °C in the open and closed systems. ZnO nanoparticles synthesized at 50, 60, and 70 °C in the open system have average particle sizes of 4.0–4.5 nm, 4.5–5.0 nm, and 6.0–7.0 nm, respectively. On the other hand, ZnO nanoparticles synthesized in the closed system have average particle sizes of 3.0–3.5, 3.5–4.5, and 5.0–6.0 nm at synthesis temperatures of 50, 60, and 70 °C, respectively. These results are consistent with the TEM results of Figure 2, indicating that the particle size of the nanoparticles increases with increasing synthesis temperature, and the synthesis of ZnO nanoparticles in the open system results in larger particle sizes.

Figure 4 shows the average diameter of the ZnO nanoparticles synthesized at 50, 60, and 70 °C in the open and closed systems. For the ZnO nanoparticles synthesized in the closed system, the average diameters are 3.2, 4.0, and 5.4 nm at 50, 60, and 70 °C, respectively. The average diameter increases as the synthesis temperature increases. For the open system, the average diameters are 4.2, 4.7, and 6.5 nm at 50, 60, and 70 °C, respectively. The average diameter increases with increasing synthesis temperature as in the case of the close system. In addition, the ZnO nanoparticles synthesized in the open system are larger compared with those in the closed system at the same synthesis temperature. The methanol solvent evaporates as the synthesis proceeds in the open system. For example, the volume of methanol solvent is 190 mL at the starting point of the synthesis. However, the volume decreases to 110 mL after finishing the synthesis due to the evaporation of the methanol solvent. Hence, the concentration of the reacting particles increases due to the solvent evaporation, resulting in particle growth via Ostwald ripening [33]. Therefore, the particle size of ZnO nanoparticles synthesized in the open system is larger compared to that in the closed system. These results also suggest that the ZnO nanoparticles through by a diffusion-limited process [33,34].

Figure 5 shows the photoluminescence spectra for the ZnO nanoparticles synthesized at 50, 60, and 70 °C in the open and closed systems. The synthesized ZnO nanoparticles exhibit broad emission peaks in the wavelength range of 550–580 nm, attributed to radiative recombination from the surface defects of the nanoparticles [35]. ZnO nanoparticles synthesized at 50 °C in the open and closed systems exhibit emission peaks at about 560 and 555 nm, respectively. On the other hand, the peak wavelength shifts to 580 nm as the synthesis temperature increases to 70 °C in the open system. Similarly, the emission peak shifts to 575 nm upon increasing the synthesis temperature to 70 °C in the closed system. This red shift in the wavelength at higher synthesis temperatures is attributed to the increased particle size, as shown in Figure 2, Figure 3 and Figure 4.

It is known that the emission wavelength of ZnO nanoparticles in the visible range depends on the bandgap and defect levels [12,35]. The bandgap of ZnO nanoparticles decreases with increasing particle size due to the quantum confinement effect, resulting in a red-shift emission [12,35]. In addition, Figure 5 exhibits that the emission intensity decreases as the synthesis temperature increases. The photoluminescence intensity of ZnO in the visible range is dependent on the number of surface defects [34,35]. As the particle size increases with increasing synthesis temperature, as shown in Figure 2, Figure 3 and Figure 4, the surface-to-volume ratio increases so that the number of surface defects decreases. Therefore, the lower photoluminescence intensity at higher temperatures reflects the larger particle size of ZnO nanoparticles.

Inverted QLEDs were fabricated using the ZnO nanoparticles synthesized in the open and closed systems. The inverted QLEDs have a structure of ITO/ZnO nanoparticles (30 nm)/QDs (15 nm)/CBP (30 nm)/MoO_3_ (10 nm)/Al. For the emission layer of the inverted QLEDs, we used red-emitting CdSe/ZnS QDs purchased from Global ZEUS (ZEUS Co., Ltd., Hwaseong, Republic of Korea). Oleic acid ligands were used for the CdSe/ZnS QDs. These red QDs were dispersed in octane solvents. Figure 6 shows the TEM image, particle size distribution, and photoluminescence spectrum of red QDs. The TEM image shows that the QDs have a polygonal shape. The red QDs have average particle sizes of 9–12 nm. The average particle size of the red QDs is 10.3 nm. The red QDs exhibit an emission peak at 629 nm and a full width at half-maximum (FWHM) of 39 nm. The photoluminescence quantum yield of the red QDs is 90%.

Figure 7a presents the electroluminescence spectra for the inverted QLEDs with the ZnO nanoparticles synthesized in the open system. ZnO nanoparticles synthesized at 50 °C exhibit a strong emission peak at 630 nm, coinciding with the photoluminescence peak of red QDs shown in Figure 6c. This result indicates that the intensity peak at 630 nm results from the emission of CdSe/ZnS QDs. ZnO nanoparticles synthesized at 60 and 70 °C in the open system also exhibit the emission from CdSe/ZnS QDs at 628 and 630 nm, respectively. An FWHM of 40 nm is observed in the device with ZnO nanoparticles synthesized at 50 °C. The FWHMs in the devices with the ZnO synthesized at 60 and 70 °C are 35 and 36 nm, respectively. In addition, a weak emission around 400–410 nm can be observed in the electroluminescence spectrum. This emission means that parasitic recombination occurs in the CBP hole transport layer through leaking of the electrons into the CBP layer from the QD layer. The CBP emission increases with increasing synthesis temperature, as shown in Figure 7a. This result indicates that more electrons are injected into the CBP layer from the QD layer as the ZnO synthesis temperature increases. The electroluminescence spectra of Figure 7b for the inverted QLEDs with the ZnO nanoparticles synthesized in the closed system present similar results to those in Figure 7a. Strong emissions at 628–629 nm can be observed, and these red emissions have an FWHM of 36–39 nm. This figure also indicates a weak emission around 400–410 nm resulting from the radiative recombination in the CBP layer. This CBP emission due to the leakage of electrons from QD layer slightly increases with increasing synthesis temperature. However, the CBP emission is weaker in the QLEDs with the ZnO nanoparticles synthesized in the closed system.

Figure 8 shows the current density—voltage—luminance curves and the current efficiency curves for the inverted devices with the ZnO nanoparticles synthesized at 50, 60, and 70 °C in the open system. The current density is higher at higher synthesis temperature. For example, the device with the ZnO nanoparticles synthesized at 50 °C exhibits 45 mA/cm^2^ at 5 V, while the current density increases to 403 mA/cm^2^ at 5 V in the device with the ZnO nanoparticles synthesized at 70 °C. Since the average particle size increases with increasing synthesis temperature, a larger particle size at a higher synthesis temperature results in a smaller number of defects and enhances the electron conduction of ZnO nanoparticles. The required voltage to achieve the same brightness is also lower at a higher synthesis temperature. For example, the turn-on voltage, which is defined as the voltage for a luminance of 1 cd/m^2^, decreases from 2.8 to 2.4 V as the synthesis temperature of ZnO nanoparticles increases from 50 to 70 °C. The driving voltage for 1000 cd/m^2^ also decreases from 6.4 to 4.4 V upon increasing the synthesis temperature from 50 to 70 °C. A luminance of 3490 cd/m^2^ is achieved at a voltage of 8 V for the device with the ZnO nanoparticles synthesized at 50 °C, while the luminance reaches 8250 cd/m^2^ at the same voltage for the device with 70 °C nanoparticles. The higher luminance at a higher synthesis temperature results from the current density rather than the current efficiency. In the luminance range of 1–300 cd/m^2^, the current efficiency is higher in the device with the ZnO nanoparticles synthesized at a lower temperature. For example, the current efficiencies at about 150 cd/m^2^ are 0.15, 0.24, and 0.34 cd/A at synthesis temperatures of 50, 60, and 70 °C, respectively. Since a higher current density is obtained at a higher synthesis temperature, as shown in Figure 8a, a higher electron current at a higher synthesis temperature makes the charge balance worse. However, the current efficiency is higher in a 60 °C device in the luminance range of 2000–5000 cd/m^2^. For example, the current efficiencies are 0.36, 0.75, and 0.46 cd/A at about 3000 cd/m^2^ for the devices with ZnO nanoparticles synthesized at 50, 60, and 70 °C, respectively, in the open system. This result indicates that the charge balance is dependent on the electric field distribution in the device rather than the electron current.

Figure 9 shows the current density—volage—luminance curves and the current efficiency curves for the inverted devices with the ZnO nanoparticles synthesized at 50, 60, and 70 °C in the closed system. The device with the ZnO nanoparticles synthesized in the closed system exhibits similar results to the device in the open system. The higher synthesis temperature results in a higher current density at the same voltage because of the larger particle size. The current densities at 5 V are 34 and 322 mA/cm^2^ at 50 and 70 °C, respectively. The current density is slightly lower in the closed system compared to the open system. However, the device with nanoparticles synthesized in the closed system exhibits lower driving volage compared to the open system. For example, the turn-on voltage of the devices at 50 °C is 2.6 in the closed system, while the turn-on voltage is 2.8 V at 50 °C in the open system. The voltage for the same luminance is higher in the device with the ZnO nanoparticles synthesized at 50 °C in the closed system. For example, the turn-on voltage decreases from 2.6 to 2.2 V as the synthesis temperature increases from 50 to 70 °C. The voltage for a luminance of 1000 cd/m^2^ also decreases from 5.4 to 3.4 V with increasing synthesis temperature from 50 to 70 °C. The higher current density at the higher synthesis temperature results in higher luminance. On the other hand, the current efficiency is higher in the device with the ZnO nanoparticles synthesized at 60 °C in the closed system, as shown in Figure 9b. The 60 °C device exhibits the maximum current efficiency of 5.8 cd/A attributed to the better charge balance, while the maximum efficiencies are 2.39 and 3.02 cd/A in the devices with the nanoparticles synthesized at 50 and 70 °C, respectively. In addition, the maximum current efficiency is higher in the closed system compared to the open system, suggesting that the charge balance can be more optimized by using the ZnO nanoparticles synthesized in the closed system.

## 4. Conclusions

We synthesized wurtzite-structured ZnO nanoparticles using potassium hydroxide and zinc acetate dihydrate at 50, 60, and 70 °C in open and closed systems to produce an electron transport layer of inverted red QLEDs. Upon increasing the synthesis temperature from 50 to 70 °C, the particle size distribution shifted toward higher particle sizes, and the average particle size increased. The synthesized nanoparticles exhibited photoluminescence emission in the visible wavelength range, resulting from the surface defects of the nanoparticles. The photoluminescence spectrum was shifted toward the longer wavelength and the emission intensity decreased as the synthesis temperature increased from 50 to 70 °C. The average particle size was larger and the visible emission was weaker in the open system compared to the closed system since the solvent evaporated during the synthesis process in the open system. The driving voltage of the inverted QLEDs decreased with increasing synthesis temperature, which was attributed to the increased particle size and the reduced number of surface defects. The inverted red QLEDs with ZnO nanoparticles synthesized at 60 °C in the closed system exhibited a maximum current efficiency of 5.8 cd/A.

## Figures and Tables

**Figure 1 micromachines-15-01297-f001:**
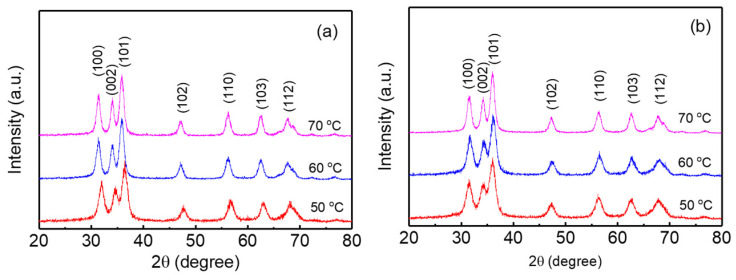
The X-Ray diffraction patterns for the ZnO nanoparticles synthesized at various temperatures in the (**a**) open and (**b**) closed systems.

**Figure 2 micromachines-15-01297-f002:**
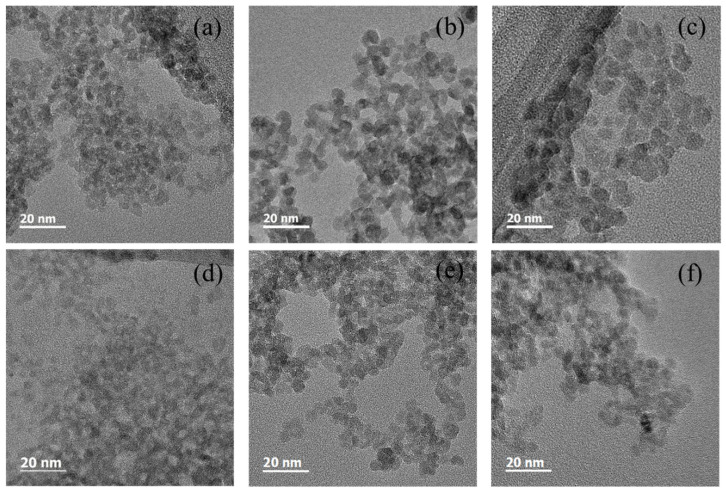
TEM images for the ZnO nanoparticles synthesized at (**a**) 50, (**b**) 60, and (**c**) 70 °C in the open system and at (**d**) 50, (**e**) 60, and (**f**) 70 °C in the closed system.

**Figure 3 micromachines-15-01297-f003:**
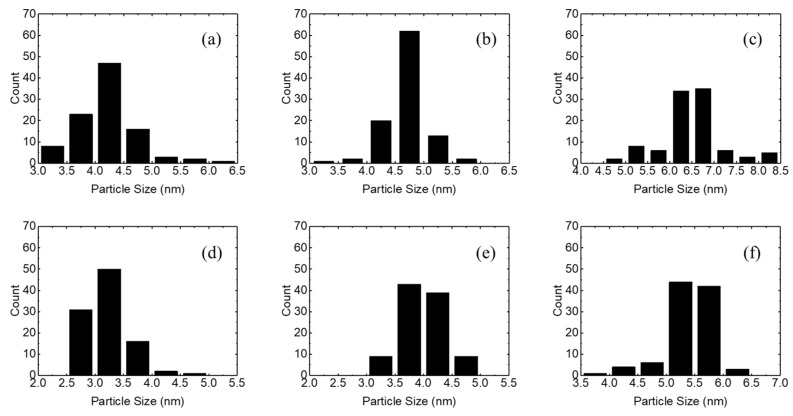
Particle size distributions for the ZnO nanoparticles synthesized at (**a**) 50, (**b**) 60, and (**c**) 70 °C in the open system and at (**d**) 50, (**e**) 60, and (**f**) 70 °C in the closed system.

**Figure 4 micromachines-15-01297-f004:**
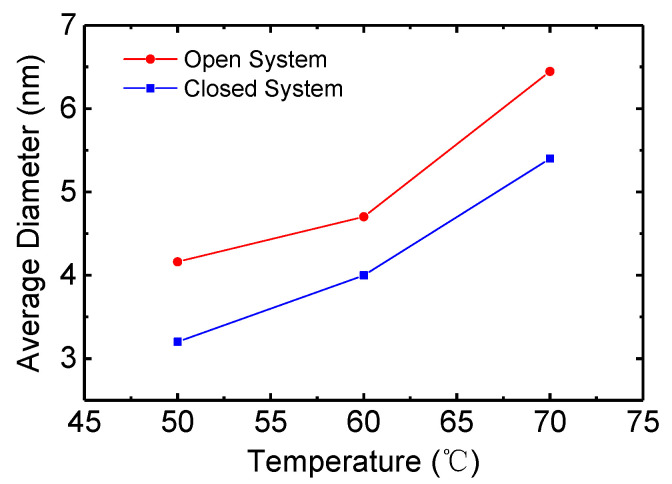
The average diameters of the ZnO nanoparticles synthesized at various temperatures in the open and closed systems.

**Figure 5 micromachines-15-01297-f005:**
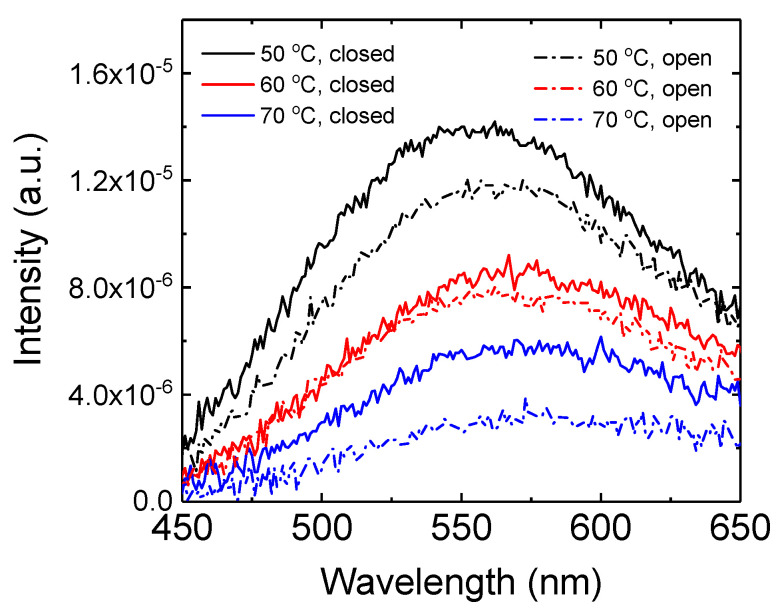
Photoluminescence spectra for the ZnO nanoparticles synthesized at various temperatures in the open and closed systems.

**Figure 6 micromachines-15-01297-f006:**
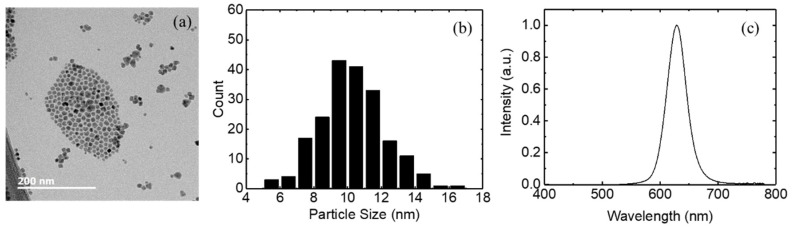
The (**a**) TEM image, (**b**) particle size distribution, and (**c**) photoluminescence spectrum of the red QDs.

**Figure 7 micromachines-15-01297-f007:**
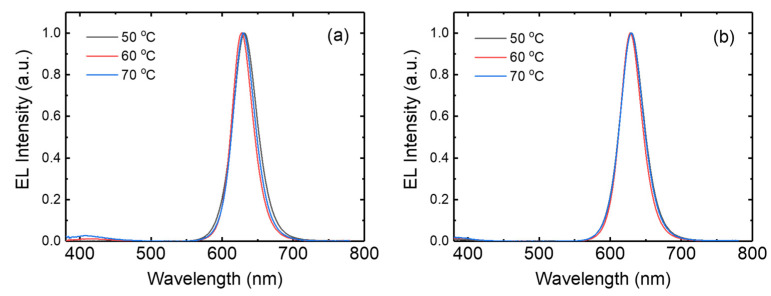
Electroluminescence spectra for the QLEDs with ZnO nanoparticles synthesized at various temperatures in the (**a**) open and (**b**) closed systems.

**Figure 8 micromachines-15-01297-f008:**
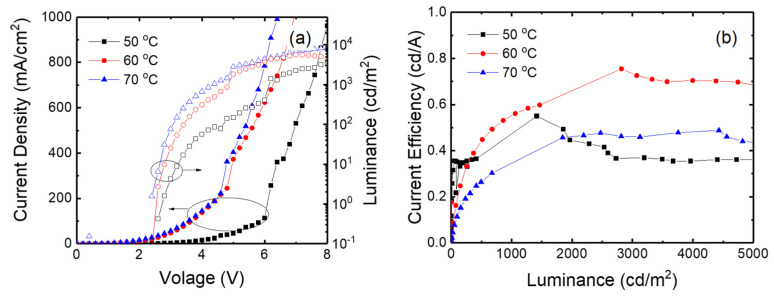
(**a**) Current density—voltage—luminance curves and (**b**) current efficiency curves for the inverted devices with the ZnO nanoparticles synthesized at various temperatures in the open system.

**Figure 9 micromachines-15-01297-f009:**
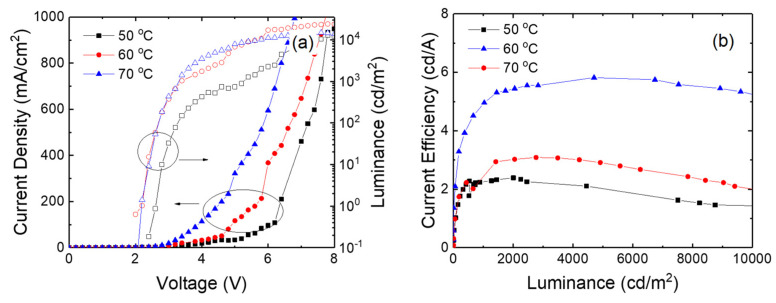
(**a**) Current density—volage—luminance curves and (**b**) current efficiency curves for the inverted devices with the ZnO nanoparticles synthesized at various temperatures in the closed system.

## Data Availability

The data are contained within the article.

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
