# Peer review of "Inverted Red Quantum Dot Light-Emitting Diodes with ZnO Nanoparticles Synthesized Using Zinc Acetate Dihydrate and Potassium Hydroxide in Open and Closed Systems"

_micromachines, 2024, doi:10.3390/mi15111297_

Round 1
Reviewer 1 Report
Comments and Suggestions for Authors
Dear Authors, investigating properties of ZnO particles, synthesized under different conditions, with respect to QLED emission is new and interesting. The paper is well written. The methods and materials part is very detailed, what I like, since it helps to support further work in this field by other groups. But I have some doubts about the open synthesis approach, since changing of reaction conditions has a major impact on nearly all nanostructures. Usually, one is trying to maintain constant conditions as good as possible. This is also confirmed by your results - the performance of the particles from the closed synthesis is much better. But the results are valid, and the comparison of the influence of different particle sizes is very helpful for device optimization. In future papers, it would be nice to have some more interpretation of the QLED device characteristics. But that shouldn't deminish the work presented here.
Author Response
Response to Reviewer 1 Comments:
Thank you very much for taking the time to review this manuscript. We revised the manuscript as reviewer’s comments and suggestions. Please find the detailed responses below and the corresponding revisions in the re-submitted files.
Reviewer #1: Dear Authors, investigating properties of ZnO particles, synthesized under different conditions, with respect to QLED emission is new and interesting. The paper is well written. The methods and materials part is very detailed, what I like, since it helps to support further work in this field by other groups. But I have some doubts about the open synthesis approach, since changing of reaction conditions has a major impact on nearly all nanostructures. Usually, one is trying to maintain constant conditions as good as possible. This is also confirmed by your results - the performance of the particles from the closed synthesis is much better. But the results are valid, and the comparison of the influence of different particle sizes is very helpful for device optimization. In future papers, it would be nice to have some more interpretation of the QLED device characteristics. But that shouldn't deminish the work presented here.
Response: Thank you for pointing out the reaction conditions and more interpretation of the QLED device characteristics. As the reviewer pointed out, the device performance was better in the device with nanoparticles from the closed system. And we added several sentences describing the QLED device characteristics in the results and discussion section of the manuscript.
The manuscript has been substantially revised by taking care of all issues raised by the reviewer.

Reviewer 2 Report
Comments and Suggestions for Authors
Major revisions

Author Response
Response to Reviewer 2 Comments:
Thank you very much for taking the time to review this manuscript. We revised the manuscript as reviewer’s comments and suggestions. Please find the detailed responses below and the corresponding revisions in the re-submitted files.
Reviewer #2: The author compared inverted red QLEDs with ZnO nanoparticles prepared in the closed and open systems. The work can be accepted after major revisions.
Comments 1: The properties of red QDs should be added in the work, such as PL, QY (quantum yield), These informations are important to compare and analyze EL performance.
Response 1: Thank you for pointing this out. According to reviewer’s comment, we added PL spectrum and TEM photograph of red QDs. And we added the sentences describing the emission peak, FWHM, TEM particle size, PLQY of red QDs in the results and discussion section of the manuscript.
Comments 2: Size distribution of ZnO nanoparticles, such as scatter distribution, should be provided to readers. The average diameter in Figure 3 is not enough to explain the real size distribution.
Response 2: Thank you for pointing out the size distribution of ZnO nanoparticles. According to reviewer’s comment, we added the particle size distributions for the ZnO nanoparticles synthesized at 50, 60, and 70 oC in the open and closed systems.
Comments 3: The discussion needs to be more detailed. How the synthesized temperature influences the EL performance, such as luminance, EQE and J-V.
Response 3: Thank you for suggesting the detailed discussion on the effect of synthesized temperature on the EL performance. According to reviewer’s suggestion, we added several sentences describing the effect of synthesized temperature on the EL performance in the results and discussion section of the manuscript.
The manuscript has been substantially revised by taking care of all issues raised by the reviewer.

Round 2
Reviewer 2 Report
Comments and Suggestions for Authors
Accept